# Viscoelastic and Deformation Characteristics of Structurally Different Commercial Topical Systems

**DOI:** 10.3390/pharmaceutics13091351

**Published:** 2021-08-27

**Authors:** Maryam Dabbaghi, Sarika Namjoshi, Bhavesh Panchal, Jeffrey E. Grice, Sangeeta Prakash, Michael Stephen Roberts, Yousuf Mohammed

**Affiliations:** 1Therapeutics Research Group, The University of Queensland Diamantina Institute, Faculty of Medicine, University of Queensland, Brisbane, QLD 4102, Australia; m.dabbaghi@uq.edu.au (M.D.); s.namjoshi@uq.edu.au (S.N.); b.panchal@uq.edu.au (B.P.); jeff.grice@uq.edu.au (J.E.G.); m.roberts@uq.edu.au (M.S.R.); 2School of Agriculture and Food Sciences, The University of Queensland, Brisbane, QLD 4072, Australia; s.prakash@uq.edu.au; 3UniSA Clinical and Health Sciences, University of South Australia, Adelaide, SA 5001, Australia; 4Therapeutics Research Centre, Basil Hetzel Institute for Translational Health Research, The Queen Elizabeth Hospital, Adelaide, SA 5011, Australia

**Keywords:** topical semisolid products, rheology, viscoelastic properties, deformation characteristics, power-law functions, critical quality attributes (CQAs)

## Abstract

Rheological characteristics and shear response have potential implication in defining the pharmaceutical equivalence, therapeutic equivalence, and perceptive equivalence of commercial topical products. Three creams (C1 and C3 as oil-in-water and C2 as water-in-oil emulsions), and two gels (G1 and G2 carbomer-based) were characterized using the dynamic range of controlled shear in steady-state flow and oscillatory modes. All products, other than C3, met the Critical Quality Attribute criteria for high zero-shear viscosity (*η*_0_) of 2.6 × 10^4^ to 1.5 × 10^5^ Pa∙s and yield stress (*τ*_0_) of 55 to 277 Pa. C3 exhibited a smaller linear viscoelastic region and lower *η*_0_ (2547 Pa∙s) and *τ*_0_ (2 Pa), consistent with lotion-like behavior. All dose forms showed viscoelastic solid behavior having a storage modulus (G′) higher than the loss modulus (G″) in the linear viscoelastic region. However, the transition of G′ > G″ to G″ > G′ during the continual strain increment was more rapid for the creams, elucidating a relatively brittle deformation, whereas these transitions in gels were more prolonged, consistent with a gradual disentanglement of the polymer network. In conclusion, these analyses not only ensure quality and stability, but also enable the microstructure to be characterized as being flexible (gels) or inelastic (creams).

## 1. Introduction

Topically administered pharmaceutical drug products are available in solution, suspension, lotion, cream, gel, ointment, paste, and powder dosage forms. Creams, gels, and ointments are the most used forms and have a semisolid consistency [1,2], but vary in their microstructure and composition [2,3]. Furthermore, depending on the excipients and manufacturing processes used, their microstructure, resulting stability, and rheological behavior can vary significantly [3,4]. Their rheology is best characterized by changes in flow and deformation under varying shear conditions. In generic product development, rheology plays a critical role in ensuring that product quality and stability are maintained. Rheological tests are now a key component in product-specific guidances and are especially pertinent in characterizing generic drug products as being pharmaceutically equivalent to a Reference Listed Drug (RLD) product because they meet the necessary critical quality attributes [5,6,7]. Accordingly, the rheological properties of these products are regularly assessed in post-manufacturing quality assurance [8,9]. Material properties evaluated through product rheology are also an important determinant of evaporation dynamics (endothermic evaporative cooling) and the uniform distribution of an active pharmaceutical ingredient within the product, as required for therapeutic equivalence [10,11,12]. In addition, instrumentally determined rheology and evaporation are surrogates for a consumer’s “feel”, for attributes such as cooling sensation, smoothness, and velvety feel. These attributes form an important component of perceptive/sensorial equivalence and are a contributor to therapeutic and clinical outcomes.

Creams are multiphase emulsion systems that typically consist of at least two immiscible liquids, water and oil phases, stabilized by amphiphilic emulsifying molecules. Depending on the relative volume fraction of both phases and the choice of emulsifier, one phase exists in the form of droplets dispersed throughout the other, forming either an oil-in-water or water-in-oil type of stable emulsion [13,14]. Distinct from creams, gels contain polymers which form a three-dimensional network of interconnected particles in the presence of the solvent mixture, providing stiffness to the product [13,15].

Shear flow sweep and oscillatory strain sweep analysis provide complementary rheological data. In the shear sweep test, the applied shear stress increases continuously with time, and the subsequent changes in viscosity are monitored as a response to structural deformation [16]. In oscillatory strain sweep, the behavior of an elastic and a viscous portion of the product structure is characterized under applied shear in oscillation mode. The elastic component of the structure stores the deformation energy, which, during the withdrawal of stress during oscillation, drives the reformation of strained structural arrangement. Such elastic behavior of the product is represented by the elastic or storage modulus (G′). On the other hand, when the deformation energy is lost or dissipated through the irreversible movement of the particles, aggregates, crystals, or other components, it is called viscous behavior of the product, characterized by viscous or loss modulus (G″).

This study was aimed at characterizing the irreversible shear breakdown, viscoelasticity, and other rheological differences that may be found within and across the five products, especially between the creams and gels. Of particular interest was to identify the key rheological parameters that can potentially distinguish the rheological behavior and its implied product performance with limited availability of compositional and microstructural information. A wide range of shear was used to define viscosity–shear rate relationships in the shear-thinning range from static to flowing state, with extrapolation to zero- and infinite-shear rates to enable meaningful characterization of microstructural rheology. In doing so, we sought to simulate product handling while dispensing it from the container through to application to the skin. 

## 2. Materials and Methods

### 2.1. Materials

Five commercial topical semisolid products (pharmaceutical), three labeled as creams (test cream C1, C2, and C3) and two products labeled as gels (test gel G1 and G2), were procured from the local market. The details of excipients present in each of these products are outlined in Table 1. Creams C1 and C3 were oil-in-water type of emulsions, whereas C2 was a water-in-oil type of emulsion. Both the gel samples were hydrophilic in nature prepared using Carbomer 940 polymer.

### 2.2. Methods

All rheological experiments were conducted using an AR-G2 rheometer (TA instrument^®^, New Castle, DE, USA) equipped with a Peltier stage for temperature control and a 40 mm diameter parallel plate geometry. The geometry and Peltier stage surfaces were covered with 180 grit sandpaper to eliminate the wall slip effect on measurements, specifically when samples are exposed to high shear rates [17]. Slippage can otherwise lead to erroneous measurement with poor reproducibility. A sufficient volume of sample was loaded on the Peltier stage, and then geometry was lowered to a 500 μm gap. The excess amount of sample protruded beyond the geometry area was carefully removed. A pre-shear of the sample was performed at 1 s^−1^ rate of shear for 60 s followed by 120 s of equilibration without any shear. Pre-shear was performed to distribute the sample homogeneously in the shear gap and equalize the individual shear history of these commercial products which is a result of their storage, transportation, and handling conditions prior to the experiment. Samples were analyzed using steady-state flow sweep and oscillatory amplitude sweep tests as detailed below. The test temperature for all the measurements was controlled to 32 °C to represent the skin temperature during application conditions.

#### 2.2.1. Shear Stress Flow Sweep

The steady-state controlled shear stress sweep test was performed to measure the apparent viscosity as a function of shear stress. The shear stress was set to increase logarithmically from 1 to 350 Pa with measurements made at 10 points per decade in log mode. All the measures were undertaken in triplicate.

#### 2.2.2. Shear Strain Amplitude Sweep

An oscillatory amplitude sweep was performed with controlled shear strain [$\gamma \left(t\right)=\gamma_{A}\cdot \sin\omega t$] keeping the angular frequency (*ω*) constant at 6.28 rad/s and varying the amplitude of strain with time [$\gamma_{A}\left(t\right)]$ from 0.01 to 1000%. The oscillation frequency corresponds to the deflection cycle of the geometry per unit time, and amplitude relates to the deflection angle. The measurement was recorded at five points per decade of logarithmic increment in amplitude. The structural response to oscillatory shear was evaluated in terms of elastic modulus (G′) and viscous modulus (G″) for each sample analyzed in triplicate. Each test and all individual replicates (three) were conducted on fresh samples from the same commercial package.

#### 2.2.3. Statistical Analysis

Statistical analysis of the data was conducted using MINITAB^®^ 17 statistical package (Minitab Co., State College, PA, USA). One-way analysis of variance (ANOVA) with Tukey’s multiple comparison test was employed to determine the significance of differences between the treatment means at *p* < 0.05.

## 3. Results and Discussion

### 3.1. Shear Stress Flow Sweep

Figure 1 shows the viscosity profiles observed for the creams and gels under controlled steady-state incremental shear stress mode conditions. All profiles have characteristic low-shear plateau, shear-thinning flow, and high-shear viscosity regions. The strain response in the low-stress region yields a high-viscosity plateau labeled as a zero-shear viscosity (*η*_0_). As the stress increases, plastic flow occurs at critical stress, which is referred to as dynamic yield stress (*τ*_0_) [18]. The *η*_0_ value was determined through extrapolation of the low-shear plateau to zero shear stress, and the *τ*_0_ value was identified from the intersection in the transition region as shown in Figure 1a [19]. During a continuous increase in the shear stress, the degree of strain exerted in the microstructure increases correspondingly, which retards the complete reformation or relaxation of the structure for a given timeframe of a respective stress point. Therefore, above a critical stress, extreme shear-thinning arises from an irreversible breakdown or an alteration of the product microstructure and results in the viscosity being reduced by several orders of magnitude. The shear history defined by the shear sweep test, thus, describes the various rheology phases a product will go through, from a more static state where the product is in a container to a high-flow state as it is being applied to the skin.

Figure 2a shows the measured viscosity flow curve that could be adequately described by a power-law relationship between viscosity (*η*) and shear rate ($\dot{\gamma }$) as defined by the Ostwald equation [20,21].
(1)$$\eta =K\times {\dot{\gamma }}^{n-1},$$
where *K* = ‘flow coefficient’ or ‘consistency index’ (Pa∙s), and *n* = ‘power-law’ or ‘fluidity’ index.

The Ostwald model, however, considers only the shear-thinning interval assuming no yield point. Hence, the constitutive modified power-law function, defined by the Carreau model was applied, which is shown in Figure 2b. The strength of this model is that it describes the complete flow profile covering the viscosity in the limits of zero ($\dot{\gamma }\to 0$) and infinite ($\dot{\gamma }\to \infty$) shear rates [22] (Figure 2b).
(2)$$\frac{\eta \left(\dot{\gamma }\right)-\eta_{\infty }}{\eta_{0}-\eta_{\infty }}=\frac{1}{{\left(1+(\lambda \cdot {\dot{\gamma )}}^{2}\right)}^{\frac{n}{2}}},$$
where *η*($\dot{\gamma }$) = apparent viscosity, *η*_0_ = zero-shear viscosity, *η*_∞_ = infinite-shear viscosity, *λ* = relaxation time, and *n* = ‘power-law’ or ‘fluidity’ index.

Table 2 summarizes the rheological properties measured for the five products, along with the parameters derived using the Ostwald and Carreau models. Both the Ostwald and the Carreau power-law functions showed a high regression coefficient (R*^2^* ≥ 0.97) (Table 2). The *η*_0_ values determined through extrapolation and model fit were also strongly correlated (Pearson correlation *r* = 0.99, *p* < 0.0005). All products had a fluidity (power-law) index, *n* < 1 (range: 0.12–0.29), consistent with shear-thinning flow.

The zero-shear viscosity (*η*_0_) for creams C1 and C2 and gels G1 and G2 was between 10^4^ and 10^5^ Pa∙s. These values are consistent with creams being multiphase emulsion systems, stabilized by surfactants wherein the ensued structural integrity is governed by interfacial forces and the presence of a lamellar gel network giving body to the product [14]. The structural integrity and the yield strength of carbomer gels are provided by crosslinked polymer chains and the resulting closely packed individual microgel particles [15]. In both cases, at a critical stress level, binding forces are deformed, leading to structural inclusions such as droplet aggregates, a lamellar gel phase in creams, and polymer chain segments in the microgel network becoming aligned in the direction of flow [19]. Usually, at this point, the effectiveness of interactive forces between the structural components is reduced to exhibit a plastic flow. Hence, the required magnitude of *τ*_0_ is directly related to the strength of the structure that must be exceeded for the material to flow appreciably. At a sufficiently high *τ*_0_, these materials can withstand container movements during handling and transportation. However, *τ*_0_ is also a function of the amount of work required to spread the product and, if too high, may lead to a significant cohesive resistance during application on the skin. Both gels G1 and G2 and cream C1 showed a *τ*_0_ between 55 and 87 Pa (*p* > 0.05). In comparison, C2 (water-in-oil emulsion) had a significantly higher *τ*_0_ (277 Pa, *p* < 0.05) and a larger *η*_0_ plateau region, indicating a highly structured network system, with a higher resistance to external deformation stresses. The observed *η*_0_ (narrow plateau region) and *τ*_0_ values of C3 cream were the lowest compared to all other samples, explaining the relatively weaker fluid-like structure of the product. Macroscopically, C3 also behaved more like a “lotion” (pourable liquid emulsion consistency exhibiting pseudoplastic flow [1]). It is also noted that the high *η*_0_ values for C1 and C2 and both gels do not necessarily correspond with a higher yield point. Indeed, both *η*_0_ and *τ*_0_ are dictated by the viscous and elastic architecture of the product responding to varying stress levels. The consistency index (K) derived through the power-law function was in line with the respective *τ*_0_ (Pearson correlation *r* = 0.956, *p* < 0.0005) (Table 2). C2 and C3 creams showed significantly higher (296 Pa∙s) and lower (11 Pa∙s) ‘K’ values, respectively. Accordingly, although C2 had a relatively lower *η*_0_ than the C1 and gels, it may be perceived as being comparatively thicker and firmer because of its higher yield strength. It is, therefore, envisaged that the constant ‘K,’ derived through the power-law relationship, may reflect macroscopic consistency as deduced by the sensory perception for products with a narrow range in *η*_0_ and *τ*_0_.

Another important parameter defining viscosity at high (close to infinitely high) shear rates [$\eta_{\infty }=\underset{\dot{\gamma }\to \infty }{\lim}\eta \left(\dot{\gamma }\right)$] is the infinite shear viscosity (*η*_∞_). Above the yield stress and at higher shear rates seen in the application phase, the product flows more readily. In this phase, the highest shear rate defines an infinite shear viscosity *η*_∞_, representing how the product may feel when spread onto the skin surface. The gels showed relatively higher (*p* < 0.05) levels of *η*_∞_ (0.18–0.19 Pa∙s) than the creams (0.03–0.12 Pa∙s), possibly reflecting polymer hydration and amount of water, bound by individual polymer segments, as well as particles, causing an increase in microviscosity [23]. This effect may be further exacerbated by a disentanglement of network and a sensory perception of being viscous and thick. In creams, shearing of the structure allows the water in the continuous external phase to be readily absorbed, resulting in the perception of creams being thinner than gels. Nevertheless, the perception of the residual film after the partial or complete evaporation of the solvent base may vary across the products depending on the proportion and material property of nonvolatile excipients in the mixture [24]. In the case of C2, a water-in-oil emulsion, the predicted *η*_∞_ is zero, which appears to be as a consequence of a limited range of employed shear compared to its high firmness and consistency that did not allow sufficient thinning. For such materials, a test can be performed at narrow gap heights (<100 μm) to extend the shear to sufficiently high rates without causing depletion from the shearing surfaces [25], where, due to variation in localized viscosity after shearing, the *η*_∞_ derived can be a useful indicator of “in-use” product behavior such as drug permeability and evaporation.

### 3.2. Shear Strain Oscillatory Amplitude Sweep

Figure 3 shows the dynamic modulus curves of topical products analyzed through oscillatory strain sweep. At low strain amplitudes, both the elastic G′(γ) and the viscous G″(γ) modulus formed a constant plateau region known as the linear viscoelastic range (LVR) of the material (Figure 3a,b). In the LVR, the stress induced in response to imposed strain occurs in proportionality and, therefore, a linear range is found. In Figure 3c,d, G′ and loss tangent (tanδ) are plotted against the stress response. The loss tangent is a ratio of the lost and the stored deformation energy (tanδ = G″/G′), illustrating the relative contribution of the elastic and viscous components of structure [20]. For all studied samples, G′ was observed to be higher than G″ in the LVR (tanδ < 1), revealing a dominating elastic character of the products at low strain range. However, the differences between the plateau G’ and G” for gels G1 and G2 (tanδ ≈ 0.07) and C1 cream (tanδ ≈ 0.1) were substantially higher, approximating to one decade, as compared to the marginal gap observed in C2 (tanδ ≈ 0.4) and C3 (tanδ ≈ 0.9) creams. For C2 cream, the shear energy dissipated (viscous contribution) was as high as the stored form (elastic nature) by the superstructure (tanδ ≈ 0.9). The observed trend explains a greater contribution of elastic components in the microstructures of G1, G2, and C1 as compared to C2 and C3 samples (Figure 3a,b). The elasticity of gels and creams has a variety of origins. The kinetic arrest of the structure occurs due to the caging effect induced by closely packed droplets in emulsions and polymer networks in the gels. The formation of a network through these aggregated particulates can sustain the stress at low amplitude, thereby retaining or reforming the original shape of the structure, giving a linear region of viscoelastic behavior [26]. Thus, topical semisolid products exhibit a certain rigidity at low shear or practically static conditions ($\dot{\gamma }$ ≤ 0.01 s^−1^). The different levels of G′ in the LVR plateau (G′*_p_*) can, therefore, arise due to the varying strength of interactive forces between the structural components causing resistance to movement or deformation. However, it is not only the binding forces between the inclusions but also the shape, size, and material property of structural entities such as crystals, dispersed droplets, air cells, and solvent mixtures that contribute to the macroscopic strength of the product [26,27]. For practical purposes, the G′_P_ can be used to represent the mechanical rigidity or stiffness of the products [19]. In creams, the order of stiffness was evident as C2 > C1 > C3 (Figure 3a,c), where C2 was significantly higher (*p* < 0.05) than the remaining products (Table 3). The viscoelastic behavior and the stiffness for gels G1 and G2 were observed to be comparable (Figure 3b,d, Table 3).

As the controlled strain is continuously ramped, G′ starts to deviate from the LVR region at a specific strain or stress value, called the limiting value of LVR. It is also referred to as the yield point since the structure starts to undergo irreversible deformation. Thereafter, the structure begins to collapse drastically, and a transition from G′ > G″ to G″ > G′ occurs. The crossover point of G′ and G″ during this transition process is referred to as flow point (G′ = G″) beyond which the product behaves as a viscoelastic liquid (tanδ > 1). The magnitude of strain exerted at the flow point was higher for gel samples (222%) and C1 cream (168%) as compared to the C2 (3%) and C3 (50%) creams. The observed pattern of flow point strain corresponded to the relative elastic contribution of the structures (Figure 3c,d). The greater elastic nature of the products G1, G2, and C1 (tanδ < 0.1) appeared to permit a substantial elongation of the structural network under continually increasing straining force before the structure was distorted to a state of viscoelastic liquid. Between the yield point and flow point (yield zone), the contribution of the elastic response while undergoing structural alteration remains higher. Thus, apart from the LVR limiting value, the intersection point in the yield zone, as shown in Figure 3d, can also be used to represent the yield stress (*τ*_0_) value [15]. The *τ*_0_ determined using this method was in the range of 1–235 Pa for creams and 91–113 Pa for gels (Table 3), strongly corresponding to that determined through shear stress sweep analysis (*r* = 0.926, *p* < 0.0005). The range of strain imposed in the yield zone, if calculated as a ratio of strain at flow point to that of yield point (transition index), can help to understand the deformation characteristics of the products. The ratio close to 1 implies flow of the product occurring within a short range of strain, illustrating a brittle breakdown as observed in the creams and specifically C2 (Table 3). We can see that the flow strain point in C2 was the lowest among all products; however, the required amount of stress to attain that strain was significantly higher (*p* < 0.05). Hence, the product structure showed stronger network formation but brittle deformation. The observed pattern of structural breakdown in C2 can correspond to the presence of petrolatum in the formulation that typically forms an intertwined network of microcrystalline lamellar sheets [28]. The gels showed a larger transition index that characterizes the gradual stretch or disentanglement of the polymer network, which is in line with the dominant elastic nature of the gel structure (tanδ < 0.1, Figure 3c,d). The disruption of the gel network can, however, vary depending on the polymer concentration. The closer packing of microgel particles, at high polymer concentrations, can lead to lower breakdown strain or transition index [29].

## 4. Conclusions

This study highlights the analysis, derivation, and interpretation of rheological parameters that can effectively distinguish the rheological behavior of topical semisolid products and their potential implied significance on different aspects of product performance. The cream and gel products characterized using shear stress sweep and oscillatory strain sweep tests demonstrated an apparent viscoelastic solid-like consistency and plastic flow behavior. The yield stress evaluated through both the flow sweep and the oscillatory tests was in good correlation and, along with the elastic plateau modulus values, showed the varying structural strength of the products, which had a direct influence on product stability and ease of application. With controlled oscillatory strain sweep, the structural transition occurring within the yield zone distinguished the breakdown of the product’s structure as being more flexible (gels) or having brittle fracture (creams). The products having dominant elastic contribution in the structure were shown to require higher levels of shear strain to deform from a viscoelastic solid to a viscoelastic liquid state. This study also showed that the dynamic rheological behavior could vary across products before, during, or after phases of deformation or application. Accordingly, in addition to the characterization of structural arrangement, the derived rheological attributes at various stages of deformation can also be used in undertaking a comparison of quality compliance, process optimization, product stability, and performance.

## Figures and Tables

**Figure 1 pharmaceutics-13-01351-f001:**
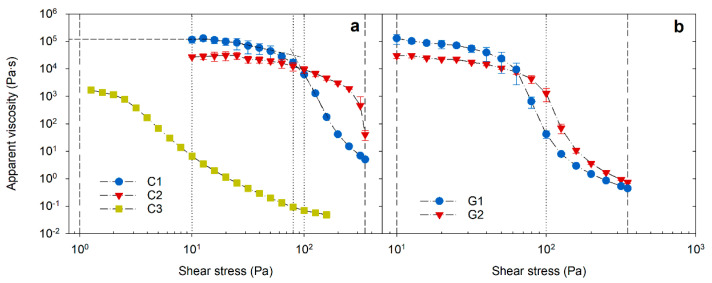
Viscosity flow sweep curves of cream (**a**) and gel (**b**) samples as a function of controlled shear stress. Reference lines from left to right are placed at 1, 10, 80, 100, and 350 Pa in ‘plot a’ and at 10, 100, and 350 Pa in ‘plot b’.

**Figure 2 pharmaceutics-13-01351-f002:**
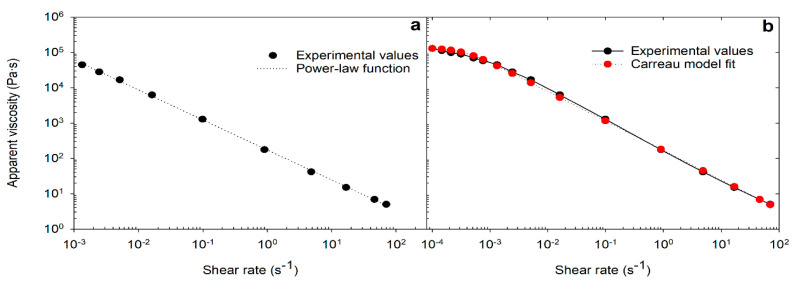
Experimental (measured) viscosity flow curves fitted to Ostwald or power-law model (**a**) and Carreau model (**b**).

**Figure 3 pharmaceutics-13-01351-f003:**
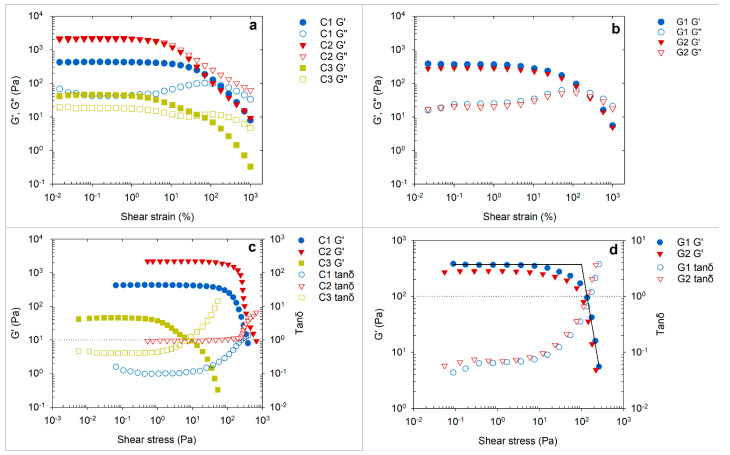
Elastic modulus (G′) and viscous modulus (G″) curves of cream (**a**) and gel (**b**) samples at controlled shear strain. G′ and loss tangent (tanδ) curves of cream (**c**) and gel (**d**) samples as a function of shear stress. Horizonal reference lines in ‘plot c’ and ‘plot d’ are placed at 1 tanδ.

**Table 1 pharmaceutics-13-01351-t001:** Excipients present in commercial topical creams (C1, C2, and C3) and gels (G1 and G2).

Products	Excipients																
	Polyoxyethylene Fatty Acid Esters	Carboxy Polymethylene	Cetostearyl Alcohol	Glycerol Monostearate	Polyoxyethylene Stearate	Poloxamer 407	Carbomer 940	Sodium Lauryl Sulfate	Disodium EDTA	White Petrolatum	White Vaseline	Dimethicone	Propylene Glycol	Mineral Oil	Preservative	Sodium Hydroxide	Purified Water
C1	✓	✓	∙	∙	∙	∙	∙	∙	∙	∙	∙	∙	∙	∙	∙	✓	✓
C2	∙	∙	✓	∙	∙	✓	∙	✓	∙	✓	∙	∙	∙	✓	∙	∙	✓
C3	∙	∙	∙	✓	✓	∙	∙	∙	∙	∙	✓	✓	∙	✓	∙	∙	✓
G1	∙	∙	∙	∙	∙	∙	✓	∙	✓	∙	∙	∙	✓	∙	✓	✓	✓
G2	∙	∙	∙	∙	∙	∙	✓	∙	✓	∙	∙	∙	✓	∙	✓	✓	✓

Symbols **✓** and · indicate the presence and absence, respectively, of an excipient in the composition.

**Table 2 pharmaceutics-13-01351-t002:** Rheological properties of cream and gel samples derived through controlled shear stress flow sweep.

Sample	Extrapolation		Carreau Model Fit				Ostwald Model Fit		
	Yield Stress	Zero Shear Viscosity	Zero Shear Viscosity	Infinite Shear Viscosity	Fluidity Index	Correlation Coefficient	Consistency Index	Fluidity Index	Correlation Coefficient
	*τ*_0_ (Pa)	*η*_0_ (Pa∙s)	*η*_0_ (Pa∙s)	*η*_∞_ (Pa∙s)	*n*	R^2^	K (Pa∙s)	n	R^2^
C1	81.45 ^b^	1.6 × 10^5 a^	1.3 × 10^5 a^	0.12 ^b^	0.17	0.98	168.09 ^b^	0.17	0.99
C2	276.7 ^a^	2.6 × 10^4 b^	2.5 × 10^4 c^	0.00 ^c^	0.20	0.99	296.1 ^a^	0.25	0.97
C3	2.27 ^c^	2547 ^b^	3708 ^d^	0.03 ^c^	0.26	0.99	10.66 ^d^	0.29	0.99
G1	54.99 ^bc^	1.3 × 10^5 a^	1.1 × 10^5 b^	0.19 ^a^	0.12	0.97	105.51 ^c^	0.15	0.99
G2	86.57 ^b^	3.1 × 10^4 b^	2.4 × 10^4 c^	0.18 ^a^	0.12	0.98	121.28 ^bc^	0.15	0.99

^a–d^ Data that do not share the same letter(s) in respective column are significantly different (*p* < 0.05).

**Table 3 pharmaceutics-13-01351-t003:** Rheological properties of cream and gel samples derived through controlled shear strain oscillatory sweep.

Sample	Plateau G′ (G′_P_) (Pa)	Yield Stress (*τ*_0_) (Pa)	Yield Point Strain (%)	Flow Point Strain (%)	Flow Transition Index
C1	432.48 ^b^	132.70 ^b^	2.46 ^a^	167.80 ^b^	68.21 ^c^
C2	2192.00 ^a^	234.70 ^a^	0.98 ^b^	2.60 ^d^	2.64 ^d^
C3	45.55 ^b^	1.19 ^d^	0.62 ^c^	50.44 ^c^	81.35 ^b^
G1	371.30 ^b^	113.00 ^bc^	0.97 ^b^	222.10 ^a^	229.00 ^a^
G2	272.90 ^b^	90.91 ^c^	0.97 ^b^	222.10 ^a^	229.00 ^a^

^a–d^ Data that do not share letter(s) in respective columns are significantly different (*p* < 0.05).

## Data Availability

The data presented in this study are available on request from the corresponding author. The data are not publicly available due to privacy restrictions.

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
