# Peer review of "Viscoelastic and Deformation Characteristics of Structurally Different Commercial Topical Systems"

_pharmaceutics, 2021, doi:10.3390/pharmaceutics13091351_

Round 1
Reviewer 1 Report
For all figures, please increase the size of symbols and use scientific tick labels (10^3 would be better than 1E3) for both x- and y-axes.
The authors selected to apply a preshear at 1 s–1 for 1 min then left the samples to relax for 2 min. On which basis have you determined these conditions (preshear and relaxation)?
For each sample, have you applied both tests (shear flow and strain amplitude) to the same sample or a fresh sample was used in each test? Why did you decide to start with the more destructive shear flow test than strain amplitude test?
The shear thinning was ascribed to the irreversible breakdown or an alteration of the product microstructure. Have you examined the reversibility by applying down-scan shear stress?
In the linear viscoelastic regimes, (Figure 3a) creams show gel-like response with G’ higher than G’’. The difference between G’ and G’’ is almost one decade for C1 but less for C3 and much less (almost no difference) for C2. Could you please explain this trend?
In the strain sweep results of C2 (Fig. 3a) and G1 and G2 (Fig. 3b), clear strain overshoots are observed, whereas C1 and C3 do not show such behavior, could you please explain?
Author Response
Reviewer 1
We would like to thank reviewer 1 for providing these extremely constructive insights. We have taken the suggestions on board and modified the manuscript wherever possible. In other instances, we have tried to provide justification for the questions and comments that reviewer 1 has raised. We hope that our changes, responses and justification are satisfactory.
Comments and Suggestions for Authors
#1: For all figures, please increase the size of symbols and use scientific tick labels (10^3 would be better than 1E3) for both x- and y-axes.
Response: Thank you very much for this observation. The figures have been revised as per the suggested improvements.
#2: The authors selected to apply a pre-shear at 1 s–1 for 1 min then left the samples to relax for 2 min. On which basis have you determined these conditions (pre-shear and relaxation)?
Response: Based on preliminary study on range of shear the products undergo during the test, the pre-shear of 1 s-1 was selected as sufficiently high to equalize existing pre-stresses in the sample that may have occurred during handling of the product before the test. The equilibration time was set to 2 mins, consistent for all the samples. In essence we have created a new shear history which is consistent on each of the products tested. We have clarified this in L118-119 (page 3) “Pre-shear was performed to distribute the sample homogeneously in the shear gap and equalize the individual shear history of these commercial products which is a result of their storage, transportation, and handling conditions prior to the experiment.”.
#3: For each sample, have you applied both tests (shear flow and strain amplitude) to the same sample or a fresh sample was used in each test? Why did you decide to start with the more destructive shear flow test than strain amplitude test?
Response: We would like to thank the reviewers for pointing to this detail. The clarification to this query has now been added in L138-139 (page 4) “Each test and all individual replicates (three) were conducted on fresh sample from the same commercial package.” We found since fresh samples were used, the choice of test and the order of testing did not have influence on the test outcome.
#4: The shear thinning was ascribed to the irreversible breakdown or an alteration of the product microstructure. Have you examined the reversibility by applying down-scan shear stress?
Response: The irreversible breakdown of the structure causing shear-thinning occurs due to the continuous increment in the shear stress beyond yield point. The time frame of each increasing shear stress point is not long enough to allow reversibility of the structure. The reversibility of the deformation could occur depending on the product structure when applied strain is gradually reduced or removed completely and the sample is allowed to relax. Based on this comment, it has been now specified in L156-159 (page 4) “During a continuous increase in the shear stress, the degree of strain exerted in the micro-structure increases correspondingly, which retards the complete reformation or relaxation of the structure for a given timeframe of a respective stress point.”.
#5: In the linear viscoelastic regimes, (Figure 3a) creams show gel-like response with G’ higher than G’’. The difference between G’ and G’’ is almost one decade for C1 but less for C3 and much less (almost no difference) for C2. Could you please explain this trend?
Response: A discussion has been added to explain this trend in L270-279 (page 8) “For all studied samples, G′ was observed to be higher than G″ in the LVR (tanδ < 1), revealing a dominating elastic character of the products at low strain range. However, the differences between the plateau G’ and G” for gels, G1 and G2 (tanδ~ 0.07) and C1 cream (tanδ~ 0.1) were substantially higher, approximating to one decade, as compared to the marginal gap observed in C2 (tanδ~ 0.4) and C3 (tanδ~ 0.9) creams. For C2 cream, the shear energy dissipated (viscous contribution) was as high as the stored form (elastic nature) by the superstructure (tanδ~ 0.9). The observed trend explains a greater contribution of elastic components in the microstructures of G1, G2, and C1 as compared to C2 and C3 samples (Figure 3a and b).”.
This observation has also been related to the part of discussion in L334-337 (page 10) “The gels showed a larger transition index that characterizes the gradual stretch or disentanglement of the polymer network which is in line with the dominant elastic nature of gel structure (tanδ < 0.1, Figure 3c and d).”.
#6: In the strain sweep results of C2 (Fig. 3a) and G1 and G2 (Fig. 3b), clear strain overshoots are observed, whereas C1 and C3 do not show such behavior, could you please explain?
Response: An explanation to address above comment has been added in L312-318 (page 10) “The magnitude of strain exerted at the flow point was higher for gel samples (222%) and C1 cream (168%) as compared to the C2 (3%) and C3 (50%) creams. The observed pattern of flow point strain corresponded to the relative elastic contribution of the structures (Figure 3c and d). The greater elastic nature of the products G1, G2, and C1 (tanδ < 0.1) appeared to permit a substantial elongation of the structural network under continually increasing straining force before the structure was distorted to a state of viscoelastic liquid.”
Reviewer 2 Report
Title: Viscoelastic and deformation characteristics of structurally different commercial topical systems
Overview and general recommendation:
I commend the authors on their timely research paper that explores how rheological parameters can be used as predictors of product performance. Even though rheology parameters are increasingly becoming key attributes within semisolid product characterization, the current study lacks novelty. I would encourage the authors to perform additional rheology studies, such as creep tests, thixotropic behaviour, which are currently requested by the regulatory agencies in what concerns semisolid characterization.
Furthermore, it would be of interest to present data comparing the microstructure properties of the studied formulations and release behaviour.
In addition to these tests, I also suggest some modifications:
- The language needs editing. It doesn't flow well, there are isolated sentences, areas with much detail others not completely discussed. This will significantly improve paper readability.
- Table 2, please regard the scientific notation;
- Line 188: “(…) qualitatively”, should read instead: macroscopically
Author Response
Responses Attached

Reviewer 3 Report
The article may be an useful contribution to the journal; however, few changes should be taken into consideration:
Abstract: prefferably use words instead of symbols (e.g. “&”)
Products should be characterised in more details. For example, lines 188-190: “Qualitatively, C3 also behaved more like a “lotion” (pourable liquid emulsion consistency exhibiting pseudoplastic flow [1]), which could be due to the higher concentration of water in the product compared to other creams” – there are is clear data regarding percentage of water or other components in tested products. More data regarding the tested products should be provided
Aim of study must be more clearly stated. Conclusions section must be shortened and present the take home message of the study.
Statistics secion should be revised to explicit more clearly if Anova test was used with Tukey post-hoc test, to eliminate all possible term confusions.
Grammar and punctuation must also be carefully checked within the entire article .
Author Response
Reviewer 3
Comments and Suggestions for Authors
The article may be an useful contribution to the journal; however, few changes should be taken into consideration:
Response: We would like to thank the reviewer for the valuable comments and inputs. We have addressed each of the reviewer’s comments below:
#1: Abstract: prefferably use words instead of symbols (e.g. “&”)
Response: The corrections have been made in abstract L19-26 (page 1) “Rheological characteristics and shear response have potential implication in defining pharmaceutical equivalence, therapeutic equivalence, and perceptive equivalence of commercial topical products. Three creams (C1 and C3 as oil-in-water and C2 as water-in-oil emulsions), and two gels (G1 and G2 carbomer based) were characterized using the dynamic range of controlled shear in steady-state flow and oscillatory modes. All products, other than C3, met the Critical Quality Attribute criteria for high zero-shear viscosity (η0) of 2.6 × 104 to 1.5 × 105 Pa∙s and yield stress (τ0) of 55 to 277 Pa. C3 exhibited smaller linear viscoelastic region and lower η0 (2547 Pa∙s) and τ0 (2 Pa), consistent with lotion-like behavior.”.
#2: Products should be characterised in more details. For example, lines 188-190: “Qualitatively, C3 also behaved more like a “lotion” (pourable liquid emulsion consistency exhibiting pseudoplastic flow [1]), which could be due to the higher concentration of water in the product compared to other creams” – there are is clear data regarding percentage of water or other components in tested products. More data regarding the tested products should be provided.
Response: The information related to the compositional profile is usually not fully available for commercial pharmaceutical products. The attribution regarding the higher amount of water was made based on visual macroscopic observation (free flowing) and in support of observed rheological behaviour of the product. The study was therefore aimed to identify the key rheological parameters that can effectively distinguish the product behaviour under applied controlled shear (simulating the application conditions) with limited availability of compositional and microstructural information. This has now been clarified in lines 82-91 (page 2) “The study was aimed to characterize the irreversible shear breakdown, viscoelasticity, and other rheological differences that may be found within and between the five products, especially between the creams and gels. Of particular interest was to identify the key rheological parameters that can potentially distinguish the rheological behaviour and its implied product performance with limited availability of compositional and microstructural information. A wide range of shear was used to define viscosity-shear rate relationships in the shear-thinning range from static to flowing state, with extrapolation to zero- and infinite-shear rates to enable meaningful characterization of microstructural rheology. In doing so, we sought to simulate product handling while dispensing it from the container through to application to the consumer skin.”
#3: Aim of study must be more clearly stated. Conclusions section must be shortened and present the take home message of the study.
Response: As stated in the response above, authors have modified the introduction to more clearly highlight the aim of the study; L82-91 (page 2) “The study was aimed to characterize the irreversible shear breakdown, viscoelasticity, and other rheological differences that may be found within and between the five products, especially between the creams and gels. Of particular interest was to identify the key rheological parameters that can potentially distinguish the rheological behaviour and its implied product performance with limited availability of compositional and microstructural information. A wide range of shear was used to define viscosity-shear rate relationships in the shear-thinning range from static to flowing state, with extrapolation to zero- and infinite-shear rates to enable meaningful characterization of microstructural rheology. In doing so, we sought to simulate product handling while dispensing it from the container through to application to the consumer skin.”
Also, as per the reviewer’s suggestion, we have shortened the conclusions section to present the take home message of the study more clearly; L344-368 (page 11) “This study highlights the analysis, derivation, and interpretation of rheological parameters that can effectively distinguish the rheological behavior of topical semisolid products and its potential implied significance on different aspects of product performance. The cream and gel products characterized using shear stress sweep and oscillatory strain sweep tests demonstrated an apparent viscoelastic solid-like consistency and plastic flow behavior. The yield stress evaluated through both the flow sweep and oscillatory tests were in good correlation and, along with the elastic plateau modulus values, showed varying structural strength of the products, which has a direct influence on product stability and ease of application. With controlled oscillatory strain sweep, the structural transition occurring within the yield zone distinguished the breakdown of the product’s structure as being more flexible (gels) or having brittle fracture (creams). The products having dominant elastic contribution in the structure showed to require higher levels of shear strain to deform from viscoelastic solid to a viscoelastic liquid state. This study also showed that the dynamic rheological behavior could vary between the products before, during, or after phases of deformation or application. Accordingly, in addition to the characterization of structural arrangement, the derived rheological attributes at various stages of deformation can also be used in undertaking a comparative quality compliance analysis, process optimization, product stability and performance.”
#4: Statistics secion should be revised to explicit more clearly if Anova test was used with Tukey post-hoc test, to eliminate all possible term confusions.
Response: We have used ANOVA with Tukey post-hoc test. The section has been simplified in L135-139 (page 4) “Statistical analysis of the data was conducted using MINITAB® 17 statistical package (Minitab Co., USA). One-way analysis of variance (ANOVA) with Tukey's multiple comparison test was employed to determine the significance of differences between the treatment means at p < 0.05.”.
#5: Grammar and punctuation must also be carefully checked within the entire article.
Response: As per the above recommendation, we have modified many sections in the document to improve flow and added more detail and explanation where it was lacking. All the changes are highlighted as track changes so that they are easy to follow.
Reviewer 4 Report
The manuscript submitted for publication deals with the evaluation of the rheological properties of structurally different commercial topical products, namely 3 creams and 2 gels, with a future aim of defining pharmaceutical equivalence, therapeutic equivalence, and perceptive/sensorial equivalence.
The research topic is relevant. The study was well-designed, with robust experimental data. The paper is well-structured, with a convincing explanation and discussion of the obtained results.
However, some minor issues must be addressed before the acceptance of its publication.
- Please give more information about the commercial topical products (2.1. Materials): are they topical medicines or cosmetic products?
- Add that gels exhibit a hydrophilic nature (line 86, page 2);
- Please put the exponent of base-ten notation superscript of the zero
shear viscosity values (Pa.s);
- In the sentence “.. representing how the product may feel when rubbed onto the skin surface.” (lines 204-205, page 6), replace “rubbed” by “spread”;
With thanks and best wishes,
The Reviewer
Author Response
Reviewer 4
Comments and Suggestions for Authors
The manuscript submitted for publication deals with the evaluation of the rheological properties of structurally different commercial topical products, namely 3 creams and 2 gels, with a future aim of defining pharmaceutical equivalence, therapeutic equivalence, and perceptive/sensorial equivalence.
The research topic is relevant. The study was well-designed, with robust experimental data. The paper is well-structured, with a convincing explanation and discussion of the obtained results.
However, some minor issues must be addressed before the acceptance of its publication.
Response: We would like to thank the reviewer for the encouraging remarks. We have addressed all the comments below:
#1: Please give more information about the commercial topical products (2.1. Materials): are they topical medicines or cosmetic products?
Response: The products tested in the present study were topical semisolid products (pharmaceutical) available commercially. They were not cosmetic products. We have now clarified this in section 2.1., Line 100-102 (page 3) “Five commercial topical semisolid products (pharmaceutical), three labeled as creams (test cream C1, C2, and C3) and two products labeled as gels (test gel G1 and G2), were procured from the local market.”
#2: Add that gels exhibit a hydrophilic nature (line 86, page 2);
Response: The detail has been added in Line 103-104 (page 3) “Both the gel samples were hydrophilic in nature prepared using the Carbomer 940 polymer.”
#3: Please put the exponent of base-ten notation superscript of the zero shear viscosity values (Pa.s);
Response: The corrections have been made in Table 2.
#4: In the sentence “..representing how the product may feel when rubbed onto the skin surface.” (lines 204-205, page 6), replace “rubbed” by “spread”.
Response: A correction has been made in L216 (page 7) “In this phase, the highest shear rate defines an infinite shear viscosity η∞, representing how the product may feel when spread onto the skin surface.”
Round 2
Reviewer 2 Report
I commend the authors on their revised version of the present manuscript. I was pleased to know that a separate publication will address both release and permeation behaviour of the studied commercial products as data on the correlation between microstructure / performance / efficacy profile of different semisolid technological systems is highly needed.
I accept the revised version of the manuscript has improved. I solely have two minor suggestions:
- Table 1. It would be interesting to add excipient function. E.g. carbomer 940 (Gelling agent).
- Table 2: Please present in the first column the zero shear viscosity values, and then in the second column the yield point.
Reviewer 3 Report
All suggested changes have been addressed by authors; article has been improved